# How Much Position Information Do Convolutional Neural Networks Encode?

**Md Amirul Islam**[*,1,2] , **Sen Jia**[*,1] , **Neil D. B. Bruce**[1,2]
[1]Ryerson University, Canada
[2]Vector Institute for Artificial Intelligence, Canada
`amirul@scs.ryerson.ca, sen.jia@ryerson.ca, bruce@ryerson.ca`

## Abstract

In contrast to fully connected networks, Convolutional Neural Networks (CNNs) achieve efficiency by learning weights associated with local filters with a finite spatial extent. An implication of this is that a filter may know what it is looking at, but not where it is positioned in the image. Information concerning absolute position is inherently useful, and it is reasonable to assume that deep CNNs may implicitly learn to encode this information if there is a means to do so. In this paper, we test this hypothesis revealing the surprising degree of absolute position information that is encoded in commonly used neural networks. A comprehensive set of experiments show the validity of this hypothesis and shed light on how and where this information is represented while offering clues to where positional information is derived from in deep CNNs.

## 1 Introduction

Convolutional Neural Networks (CNNs) have achieved state-of-the-art results in many computer vision tasks, e.g. object classification (Simonyan & Zisserman, 2014; He et al., 2016) and detection (Redmon et al., 2016; Ren et al., 2015), face recognition (Taigman et al., 2014), semantic segmentation (Long et al., 2015; Chen et al., 2018; Noh et al., 2015; Islam et al., 2017) and saliency detection (Cornia et al., 2018; Li et al., 2014; Jia & Bruce, 2019; Islam et al., 2018). However, CNNs have faced some criticism in the context of deep learning for the lack of interpretability (Lipton, 2018).

The classic CNN model is considered to be spatially-agnostic and therefore *capsule* (Sabour et al., 2017) or recurrent networks (Visin et al., 2015) have been utilized to model relative spatial relationships within learned feature layers. It is unclear if CNNs capture any absolute spatial information which is important in position-dependent tasks (e.g. semantic segmentation and salient object detection). As shown in Fig. 1, the regions determined to be most salient (Jia & Bruce, 2018) tend to be near the center of an image. While detecting saliency on a cropped version of the images, the most salient region shifts even though the visual features have not been changed. This is somewhat surprising, given the limited spatial extent of CNN filters through which the image is interpreted. In this paper, we examine the role of absolute position information by performing a series of *randomization tests* with the hypothesis that CNNs might indeed learn to encode position information as a cue for decision making. Our experiments reveal that position information is implicitly learned from the commonly used padding operation (*zero-padding*). Zero-padding is widely used for keeping the same dimensionality when applying convolution. However, its hidden effect in representation learning has been long omitted. This work helps to better understand the nature of the learned features in CNNs and highlights an important observation and fruitful direction for future investigation.

Previous works try to visualize learned feature maps to demystify how CNNs work. A simple idea is to compute losses and pass these backwards to the input space to generate a pattern image that can maximize the activation of a given unit (Hinton et al., 2006; Erhan et al., 2009). However, it is very difficult to model such relationships when the number of layers grows. Recent work (Zeiler & Fergus, 2014) presents a non-parametric method for visualization. A deconvolutional network (Zeiler et al., 2011) is leveraged to map learned features back to the input space and their

---

[*]Equal contribution

Figure 1: Sample predictions for salient regions for input images (left), and a slightly cropped version (right). Cropping results in a shift in position rightward of features relative to the centre. It is notable that this has a significant impact on output and decision of regions deemed salient despite no explicit position encoding and a modest change to position in the input.

results reveal what types of patterns a feature map actually learns. Another work (Selvaraju et al., 2017) proposes to combine pixel-level gradients with weighted class activation mapping to locate the region which maximizes class-specific activation. As an alternative to visualization strategies, an empirical study (Zhang et al., 2016) has shown that a simple network can achieve zero training loss on noisy labels. We share the similar idea of applying a *randomization test* to study the CNN learned features. However, our work differs from existing approaches in that these techniques only present interesting visualizations or understanding, but fail to shed any light on spatial relationships encoded by a CNN model.

In summary, CNNs have emerged as a way of dealing with the prohibitive number of weights that would come with a fully connected end-to-end network. A trade-off resulting from this is that kernels and their learned weights only have visibility of a small subset of the image. This would seem to imply solutions where networks rely more on cues such as texture and color rather than shape (Baker et al., 2018). Nevertheless, position information provides a powerful cue for where objects might appear in an image (e.g. birds in the sky). It is conceivable that networks might rely sufficiently on such cues that they implicitly encode spatial position along with the features they represent. It is our hypothesis that deep neural networks succeed in part by learning both *what* and *where* things are. This paper tests this hypothesis, and provides convincing evidence that CNNs do indeed rely on and learn information about spatial positioning in the image to a much greater extent than one might expect.

## 2 POSITION INFORMATION IN CNNS

CNNs naturally try to extract fine-level high spatial-frequency details (e.g. edges, texture, lines) in the early convolutional stages while at the deepest layers of encoding the network produces the richest possible category specific features representation (Simonyan & Zisserman, 2014; He et al., 2016; Badrinarayanan et al., 2017). In this paper, we propose a hypothesis that position information is implicitly encoded within the extracted feature maps and plays an important role in classifying, detecting or segmenting objects from a visual scene. We therefore aim to prove this hypothesis by predicting position information from different CNN archetypes in an end-to-end manner. In the following sections, we first introduce the problem definition followed by a brief discussion of our proposed position encoding network.

**Problem Formulation:** Given an input image $\mathcal{I}_m \in \mathbb{R}^{h \times w \times 3}$, our goal is to predict a gradient-like position information mask $\hat{f}_p \in \mathbb{R}^{h \times w}$ where each pixel value defines the absolute coordinates of an pixel from left→right or top→bottom. We generate gradient-like masks $\mathcal{G}_{pos} \in \mathbb{R}^{h \times w}$ (Sec. 2.2) for supervision in our experiments, with weights of the base CNN archetypes being fixed.

### 2.1 POSITION ENCODING NETWORK

Our *Position Encoding Network* (PosENet) (See Fig. 2) consists of two key components: a feed-forward convolutional encoder network $f_{enc}$ and a simple position encoding module, denoted as $f_{pem}$. The encoder network extracts features at different levels of abstraction, from shallower to deeper layers. The position encoding module takes multi-scale features from the encoder network as input and predicts the absolute position information at the end.

**Encoder:** We use ResNet and VGG based architectures to build encoder networks ($f_{enc}$) by removing the average pooling layer and the layer that assigns categories. As shown in Fig. 2, the

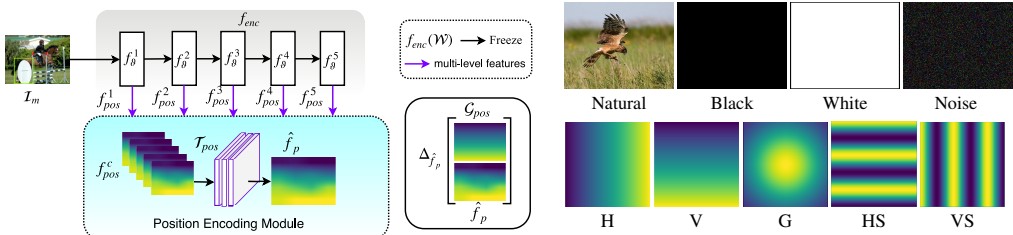

Figure 2: Illustration of PosENet architecture.

Figure 3: Sample images and generated gradient-like ground-truth position maps.

encoder module consists of five feature extractor blocks denoted by $(f_\vartheta^1, f_\vartheta^2, f_\vartheta^3, f_\vartheta^4, f_\vartheta^5)$. The extracted multi-scale features from bottom to top layers of the canonical network are denoted by $(f_{pos}^1, f_{pos}^2, f_{pos}^3, f_{pos}^4, f_{pos}^5)$. We summarize the key operations as follows:

$$f_{pos}^i = f_\vartheta^i(\mathbf{W_a} * \mathcal{I}_m) \tag{1}$$

where $\mathbf{W_a}$ denotes weights that are frozen. $*$ denotes the convolution operation. Note that in probing the encoding network, only the position encoding module $f_{pem}$ is trained to focus on extracting position information while the encoder network is forced to maintain their existing weights.

**Position Encoding Module**: The position encoding module takes multi-scale features $(f_{pos}^1, \cdots, f_{pos}^5)$ from $f_{enc}$ as input and generates the desired position map $\hat{f}_p$ thorough a transformation function $\mathcal{T}_{pos}$. The transformation function $\mathcal{T}_{pos}$ first applies a bi-linear interpolation operation on the feature maps to have the same spatial dimension resulting in a feature map $f_{pos}^c$. Once we have the same spatial dimension for multi-scale features, we concatenate them together followed by a sequence of $k \times k$ convolution operations. In our experiments, we vary the value of $k$ between $\{1, 3, 5, 7\}$ and most experiments are carried out with a single convolutional layer in the position encoding module $f_{pem}$. The key operations can be summarized as follows:

$$f_{pos}^c = (f_{pos}^1 \oplus \cdots \oplus f_{pos}^5) \quad \hat{f}_p = (\mathbf{W_{pos}^c} * f_{pos}^c) \tag{2}$$

where $\mathbf{W}_{pos}^c$ is the trainable weights attached with the transformation function $\mathcal{T}_{pos}$.

The main objective of the encoding module is to validate whether position information is implicitly learned when trained on categorical labels. Additionally, the position encoding module models the relationship between hidden position information and the gradient like ground-truth mask. The output is expected to be random if there is no position information encoded in the features maps and vice versa (ignoring any guidance from image content).

## 2.2 Synthetic Data and Ground-truth Generation

To validate the existence of position information in a network, we implement a *randomization test* by assigning a normalized gradient-like [1] position map as ground-truth shown in Fig. 3. We first generate gradient-like masks in Horizontal (**H**) and vertical (**V**) directions. Similarly, we apply a Gaussian filter to design another type of ground-truth map, Gaussian distribution (**G**) . The key motivation of generating these three patterns is to validate if the model can learn absolute position on one or two axes. Additionally, We also create two types of repeated patterns, horizontal and vertical stripes, (**HS**, **VS**). Regardless of the direction, the position information in the multi-level features is likely to be modelled through a transformation by the encoding module $f_{pem}$. Our design of gradient ground-truth can be considered as a type of random label because there is no correlation between the input image and the ground-truth with respect to position. Since the extraction of position information is independent of the content of images, we can choose any image datasets. Meanwhile, we also build synthetic images to validate our hypothesis.

## 2.3 Training the Network

As we implicitly aim to encode the position information from a pretrained network, we freeze the encoder network $f_{enc}$ in all of our experiments. Our position encoding module $f_{pem}$ generates the

---

[1]We use the term *gradient* to denote pixel intensities instead of the gradient in back propagation.

position map $\hat{f}_p$ of interest. During training, for a given input image $\mathcal{I}_m \in \mathbb{R}^{h \times w \times 3}$ and associated ground-truth position map $\mathcal{G}_{pos}^h$, we apply the supervisory signal on $\hat{f}_p$ by upsampling it to the size of $\mathcal{G}_{pos}^h$. Then, we define a pixel-wise mean squared error loss to measure the difference between predicted and ground-truth position maps as follows:

$$\Delta_{\hat{f}_p} = \frac{1}{2n} \sum_{i=1}^{n} (x_i - y_i)^2 \qquad (3)$$

where $x \in \mathbb{R}^n$ and $y \in \mathbb{R}^n$ ($n$ denotes the spatial resolution) are the vectorized predicted position and ground-truth map respectively. $x_i$ and $y_i$ refer to a pixel of $\hat{f}_p$ and $\mathcal{G}_{pos}^h$ respectively.

## 3 EXPERIMENTS

### 3.1 DATASET AND EVALUATION METRICS

**Datasets:** We use the DUT-S dataset (Wang et al., 2017) as our training set, which contains $10,533$ images for training. Following the common training protocol used in (Zhang et al., 2017; Liu et al., 2018), we train the model on the training set of DUT-S and evaluate the existence of position information on the natural images of the PASCAL-S (Li et al., 2014) dataset. The synthetic images (white, black and Gaussian noise) are also used as described in Section 2.2. Note that we follow the common setting used in saliency detection just to make sure that there is no overlap between the training and test sets. However, any images can be used in our experiments given that the position information is relatively content independent.

**Evaluation Metrics:** As position encoding measurement is a new direction, there is no universal metric. We use two different natural choices for metrics (Spearmen Correlation (SPC) and Mean Absoute Error (MAE)) to measure the position encoding performance. The SPC is defined as the Spearman's correlation between the ground-truth and the predicted position map. For ease of interpretation, we keep the SPC score within range [-1 1]. MAE is the average pixel-wise difference between the predicted position map and the ground-truth gradient position map.

### 3.2 IMPLEMENTATION DETAILS

We initialize the architecture with a network pretrained for the ImageNet classification task. The new layers in the position encoding branch are initialized with *xavier initialization* (Glorot & Bengio, 2010). We train the networks using stochastic gradient descent for 15 epochs with momentum of 0.9, and weight decay of $1e-4$. We resize each image to a fixed size of $224 \times 224$ during training and inference. Since the spatial extent of multi-level features are different, we align all the feature maps to a size of $28 \times 28$. We report experimental results for the following baselines that are described as follows: **VGG** indicates PosENet is based on the features extracted from the VGG16 model. Similarly, **ResNet** represents the combination of ResNet-152 and PosENet. **PosENet** alone denotes only the PosENet model is applied to learn position information directly from the input image. **H**, **V**, **G**, **HS** and **VS** represent the five different ground-truth patterns, horizontal and vertical gradients, 2D Gaussian distribution, horizontal and vertical stripes respectively.

### 3.3 EXISTENCE OF POSITION INFORMATION

**Position Information in Pretrained Models:** We first conduct experiments to validate the existence of position information encoded in a pretrained model. Following the same protocol, we train the VGG and ResNet based networks on each type of the ground-truth and report the experimental results in Table 1. We also report results when we only train PosENet without using any pretrained model to justify that the position information is not driven from prior knowledge of objects. Our experiments do not focus on achieving higher performance on the metrics but instead validate how much position information a CNN model encodes or how easily PosENet can extract this information. Note that, we only use one convolutional layer with a kernel size of $3 \times 3$ without any padding in the PosENet for this experiment.

As shown in Table 1, PosENet (VGG and ResNet) can easily extract position information from the pretrained CNN models, especially the ResNet based PosENet model. However, training PosENet

|   | Model | PASCAL-S SPC | PASCAL-S MAE | Black SPC | Black MAE | White SPC | White MAE | Noise SPC | Noise MAE |
|---|---|---|---|---|---|---|---|---|---|
| **H** | PosENet | .012 | .251 | .0 | .251 | .0 | .251 | .001 | .251 |
|   | VGG | .742 | .149 | .751 | .164 | .873 | .157 | .591 | .173 |
|   | ResNet | .933 | .084 | .987 | .080 | .994 | .078 | .973 | .077 |
| **V** | PosENet | .131 | .248 | .0 | .251 | .0 | .251 | .053 | .250 |
|   | VGG | .816 | .129 | .846 | .146 | .927 | .138 | .771 | .150 |
|   | ResNet | .951 | .083 | .978 | .069 | .979 | .072 | .968 | .074 |
| **G** | PosENet | -.001 | .233 | .0 | .186 | .0 | .186 | -.034 | .214 |
|   | VGG | .814 | .109 | .842 | .123 | .898 | .116 | .762 | .129 |
|   | ResNet | .936 | .070 | .953 | .068 | .964 | .064 | .971 | .055 |
| **HS** | PosENet | -.001 | .712 | -.055 | .704 | .0 | .704 | .023 | .710 |
|   | VGG | .405 | .556 | .532 | .583 | .576 | .574 | .375 | .573 |
|   | ResNet | .534 | .528 | .566 | .518 | .562 | .515 | .471 | .530 |
| **VS** | PosENet | .006 | .723 | .081 | .709 | .081 | .709 | .018 | .714 |
|   | VGG | .374 | .567 | .538 | .575 | .437 | .578 | .526 | .566 |
|   | ResNet | .520 | .537 | .574 | .523 | .593 | .514 | .523 | .545 |

Table 1: Quantitative comparison of different networks in terms of SPC and MAE across different image types.

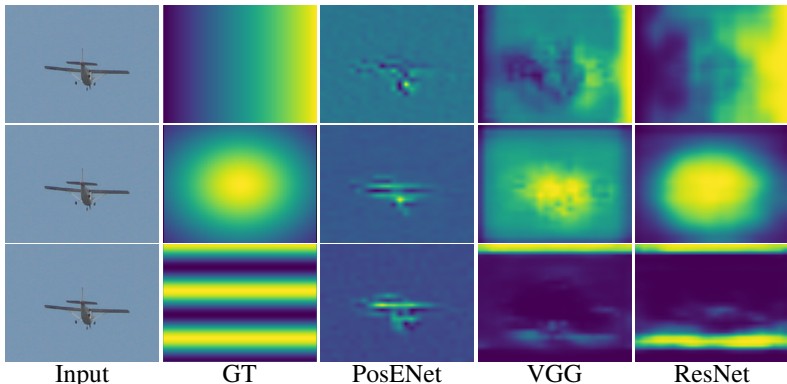

Input    GT    PosENet    VGG    ResNet

Figure 4: Qualitative results of PosENet based networks corresponding to different ground-truth patterns.

(PosENet) separately achieves much lower scores across different patterns and source images. This result implies that it is very difficult to extract position information from the input image alone. PosENet can extract position information consistent with the ground-truth position map only when coupled with a deep encoder network. As mentioned prior, the generated ground-truth map can be considered as a type of *randomization test* given that the correlation with input has been ignored (Zhang et al., 2016). Nevertheless, the high performance on the test sets across different ground-truth patterns reveals that the model is not blindly overfitting to the noise and instead is extracting true position information. However, we observe low performance on the repeated patterns (*HS* and *VS*) compared to other patterns due to the model complexity and specifically the lack of correlation between ground-truth and absolute position (last two rows of Table 1). The *H* pattern can be seen as one quarter of a sine wave whereas the striped patterns (*HS* and *VS*) can be considered as repeated periods of a sine wave which requires a deeper comprehension.

The qualitative results for several architectures across different patterns are shown in Fig. 4. We can see the correlation between the predicted and the ground-truth position maps corresponding to **H**, **G** and **HS** patterns, which further reveals the existence of position information in these networks. The quantitative and qualitative results strongly validate our hypothesis that position information is implicitly encoded in every architecture without any explicit supervision towards this objective.

Moreover, PosENet alone shows no capacity to output a gradient map based on the synthetic data. We further explore the effect of image semantics in Sec. 4.1. It is interesting to note the performance gap among different architectures specifically the ResNet based models achieve higher performance than the VGG16 based models. The reason behind this could be the use of different convolutional

| | Layers | PosENet | | VGG | |
|---|---|---|---|---|---|
| | | SPC | MAE | SPC | MAE |
| **H** | 1 Layer | .012 | .251 | .742 | .149 |
| | 2 Layers | .056 | .250 | .797 | .128 |
| | 3 Layers | .055 | .250 | .830 | .117 |
| **G** | 1 Layer | -.001 | .233 | .814 | .109 |
| | 2 Layers | .067 | .187 | .828 | .105 |
| | 3 Layers | .126 | .186 | .835 | .104 |
| **HS** | 1 Layer | -.001 | .712 | .405 | .556 |
| | 2 Layers | -.006 | .628 | .483 | .538 |
| | 3 Layers | .003 | .628 | .491 | .540 |

(a)

| | Kernel | PosENet | | VGG | |
|---|---|---|---|---|---|
| | | SPC | MAE | SPC | MAE |
| **H** | $1 \times 1$ | .013 | .251 | .542 | .196 |
| | $3 \times 3$ | .012 | .251 | .742 | .149 |
| | $7 \times 7$ | .060 | .250 | .828 | .120 |
| **G** | $1 \times 1$ | .017 | .188 | .724 | .127 |
| | $3 \times 3$ | -.001 | .233 | .814 | .109 |
| | $7 \times 7$ | .068 | .187 | .816 | .111 |
| **HS** | $1 \times 1$ | -.004 | .628 | .317 | .576 |
| | $3 \times 3$ | -.001 | .723 | .405 | .556 |
| | $7 \times 7$ | .002 | .628 | .487 | .532 |

(b)

Table 2: Quantitative comparison on the PASCAL-S dataset in terms of SPC and MAE with varying (a) number of layers and (b) kernel sizes. Note that (a) the kernel size is fixed to $3 \times 3$ but different numbers of layers are used in the PosENet. (b) Number of layers is fixed to one but we use different kernel sizes in the PosENet.

kernels in the architecture or the degree of prior knowledge of the semantic content. We show an ablation study in the next experiment for further investigation. For the rest of this paper, we only focus on the natural images, PASCAL-S dataset, and three representative patterns, **H**, **G** and **HS**.

## 3.4 ANALYZING POSENET

In this section, we conduct ablation studies to examine the role of the proposed position encoding network by highlighting two key design choices. (1) the role of varying kernel size in the position encoding module and (2) stack length of convolutional layers we add to extract position information from the multi-level features.

**Impact of Stacked Layers:** Experimental results in Table 1 show the existence of position information learned from an object classification task. In this experiment, we change the design of PosENet to examine if it is possible to extract hidden position information more accurately. The PosENet used in the prior experiment (Table 1) has only one convolutional layer with a kernel size of $3 \times 3$. Here, we apply a stack of convolutional layers of varying length to the PosENet and report the experimental results in Table 2 (a). Even though the stack size is varied, we aim to retain a relatively simple PosENet to only allow efficient readout of positional information. As shown in Table 2, we keep the kernel size fixed at $3 \times 3$ while stacking multiple layers. Applying more layers in the PosENet can improve the readout of position information for all the networks. One reason could be that stacking multiple convolutional filters allows the network to have a larger effective receptive field, for example two $3 \times 3$ convolution layers are spatially equal to one $5 \times 5$ convolution layer (Simonyan & Zisserman, 2014). An alternative possibility is that positional information may be represented in a manner that requires more than first order inference (e.g. a linear readout).

**Impact of varying Kernel Sizes:** We further validate PosENet by using only one convolutional layer with different kernel sizes and report the experimental results in Table 2 (b). From Table 2 (b), we can see that the larger kernel sizes are likely to capture more position information compared to smaller sizes. This finding implies that the position information may be distributed spatially within layers and in feature space as a larger receptive field can better resolve position information.

We further show the visual impact of varying number of layers and kernel sizes to learn position information in Fig. 5.

## 3.5 WHERE IS THE POSITION INFORMATION STORED?

Our previous experiments reveal that the position information is encoded in a pretrained CNN model. It is also interesting to see whether position information is equally distributed across the layers. In this experiment, we train PosENet on each of the extracted features, $f_{pos}^1$, $f_{pos}^2$, $f_{pos}^3$, $f_{pos}^4$, $f_{pos}^5$ separately using VGG16 to examine which layer encodes more position information. Similar to Sec. 3.3, we only apply one $3 \times 3$ kernel in $F_{pem}$ to obtain the position map.

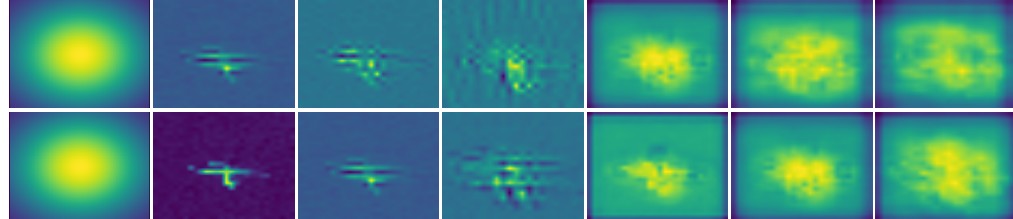

Figure 5: The effect of more **L**ayers (Top row) and varying **K**ernel **S**ize (bottom row) applied in the PoseNet. Order (left → right): GT (**G**), PoseNet (L=1, KS=1), PoseNet (L=2, KS=3), PoseNet (L=3, KS=7), VGG (L=1, KS=1), VGG (L=2, KS=3), VGG (L=3, KS=7).

| | Method | $f_{pos}^1$ | $f_{pos}^2$ | $f_{pos}^3$ | $f_{pos}^4$ | $f_{pos}^5$ | SPC | MAE |
|---|---|---|---|---|---|---|---|---|
| **H** | **VGG** | ✓ | | | | | .101 | .249 |
| | | | ✓ | | | | .344 | .225 |
| | | | | ✓ | | | .472 | .203 |
| | | | | | ✓ | | .610 | .181 |
| | | | | | | ✓ | .657 | .177 |
| | | ✓ | ✓ | ✓ | ✓ | ✓ | .742 | .149 |
| **G** | **VGG** | ✓ | | | | | .241 | .182 |
| | | | ✓ | | | | .404 | .168 |
| | | | | ✓ | | | .588 | .146 |
| | | | | | ✓ | | .653 | .138 |
| | | | | | | ✓ | .693 | .135 |
| | | ✓ | ✓ | ✓ | ✓ | ✓ | .814 | .109 |

Table 3: Performance of VGG on natural images with a varying extent of the reach of different feed-forward blocks.

As shown in Table 3, the VGG based PoseNet with top $f_{pos}^5$ features achieves higher performance compared to the bottom $f_{pos}^1$ features. This may partially a result of more feature maps being extracted from deeper as opposed to shallower layers, $512$ vs $64$ respectively. However, it is likely indicative of stronger encoding of the positional information in the deepest layers of the network where this information is shared by high-level semantics. We further investigate this effect for VGG16 where the top two layers ($f_{pos}^4$ and $f_{pos}^5$) have the same number of features. More interestingly, $f_{pos}^5$ achieves better results than $f_{pos}^4$. This comparison suggests that the deeper feature contains more position information, which validates the common belief that top level visual features are associated with global features.

## 4    WHERE DOES POSITION INFORMATION COME FROM?

We believe that the padding near the border delivers position information to learn. Zero-padding is widely used in convolutional layers to maintain the same spatial dimensions for the input and output, with a number of zeros added at the beginning and at the end of both axes, horizontal and vertical. To validate this, we remove all the padding mechanisms implemented within VGG16 but still initialize the model with the ImageNet pretrained weights. Note that we perform this experiment only using VGG based PoseNet since removing padding on ResNet models will lead to inconsistent sizes of skip connections. We first test the effect of zero-padding used in VGG, no padding used in PoseNet. As we can see from Table 4, the VGG16 model without zero-padding achieves much lower performance than the default setting (padding=1) on the natural images. Similarly, we introduce position information to the PoseNet by applying zero-padding. PoseNet with *padding*=1 (concatenating one zero around the frame) achieves higher performance than the original (*padding*=0). When we set *padding*=2, the role of position information is more obvious. This also validates our experiment in Section 3.3, that shows PoseNet is unable to extract noticeable position information because no padding was applied, and the information is encoded from a pretrained CNN model. This is why we did not apply zero-padding in PoseNet in our previous experiments. Moreover, we aim to explore

| Model | H | | G | | HS | |
|-------|-----|-----|-----|-----|-----|-----|
| | SPC | MAE | SPC | MAE | SPC | MAE |
| PosENet | .012 | .251 | -.001 | .233 | -.001 | .712 |
| PosENet with *padding*=1 | .274 | .239 | .205 | .184 | .148 | .608 |
| PosENet with *padding*=2 | .397 | .223 | .380 | .177 | .214 | .595 |
| VGG16 | .742 | .149 | .814 | .109 | .405 | .556 |
| VGG16 w/o. *padding* | .381 | .223 | .359 | .174 | .011 | .628 |

Table 4: Quantitative comparison subject to padding in the convolution layers used in PosENet and VGG (w/o and with zero padding) on natural images.

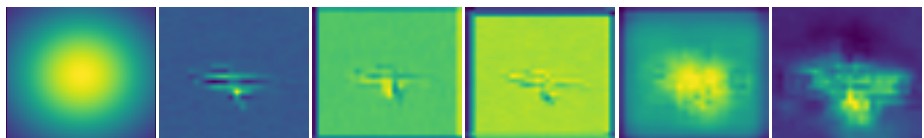

Figure 6: The effect of zero-padding on Gaussian pattern. Left to right: GT (**G**), Pad=0 (.286, .186), Pad=1 (.227, .180), Pad=2 (.473, .169), VGG Pad=1 (.928, .085), VGG Pad=0(.405, .170).

how much position information is encoded in the pretrained model instead of directly combining with the PosENet. Fig. 6 illustrates the impact of zero-padding on encoding position information subject to padding using a Gaussian pattern.

## 4.1 CASE STUDY

Recall that the position information is considered to be content independent but our results in Table 1 show that semantics within an image may affect the position map. To visualize the impact of semantics, we compute the content loss heat map using the following equation:

$$\mathcal{L} = \frac{|(\mathcal{G}_{pos}^h - \hat{f}_p^h)| + |(\mathcal{G}_{pos}^v - \hat{f}_p^v)| + |(\mathcal{G}_{pos}^g - \hat{f}_p^g)|}{3} \tag{4}$$

where $\hat{f}_p^h$, $\hat{f}_p^v$, and $\hat{f}_p^g$ are the predicted position maps from horizontal, vertical and Gaussian patterns respectively.

As shown in Figure 7, the heatmaps of PosENet have larger content loss around the corners. While the loss maps of VGG and ResNet correlate more with the semantic content. Especially for ResNet, the deeper understanding of semantic content leads to a stronger interference in generating a smooth gradient. The highest losses are from the face, person, cat, airplane and vase respectively (from left to right). This visualization can be an alternative method to show which regions a model focuses on, especially in the case of ResNet.

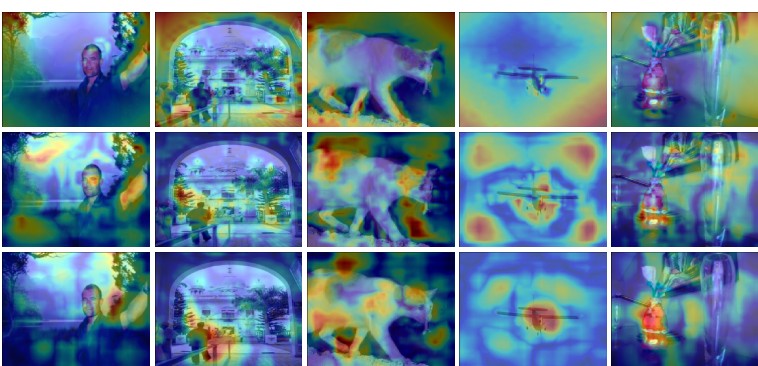

Figure 7: Error heat maps of PosENet (1st row), VGG (2nd row) and ResNet (3rd row).

## 4.2 Zero-Padding Driven Position Information

**Saliency Detection:** We further validate our findings in the position-dependent tasks (semantic segmentation and salient object detection (SOD)). First, we train the VGG network with and without zero-padding from scratch to validate if the position information delivered by zero-padding is critical for detecting salient regions. For these experiments, we use the publicly available MSRA dataset (Cheng et al., 2015) as our SOD training set and evaluate on three other datasets (ECSSD, PASCAL-S, and DUT-OMRON). From Table 5 (a), we can see that VGG without padding achieves much worse results on both of the metrics (F-measure and MAE) which further validates our findings that zero-padding is the key source of position information.

**Semantic Segmentation:** We also validate the impact of zero-padding on the semantic segmentation task. We train the VGG16 network with and without zero padding on the training set of PASCAL VOC 2012 dataset and evaluate on the validation set. Similar to SOD, the model with zero padding significantly outperforms the model with no padding.

| Model | ECSSD | | PASCAL-S | | DUT-OMRON | |
|---|---|---|---|---|---|---|
| | Fm | MAE | Fm | MAE | Fm | MAE |
| VGG w/o padding | .36 | .48 | .32 | .48 | .25 | .48 |
| VGG | .78 | .17 | .66 | .21 | .63 | .18 |

(a)

| Model | mIoU (%) |
|---|---|
| VGG w/o padding | 12.3 |
| VGG | 23.1 |

(b)

Table 5: VGG models with and w/o zero-padding for (a) SOD and (b) semantic segmentation.

We believe that CNN models pretrained on these two tasks can learn more position information than classification task. To validate this hypothesis, we take the VGG model pretrained on ImageNet as our baseline. Meanwhile, we train two VGG models on the tasks of semantic segmentation and saliency detection from scratch, denoted as VGG-SS and VGG-SOD respectively. Then we finetune these three VGG models following the protocol used in Section 3.3. From Table 6, we can see that the VGG-SS and VGG-SOD models outperform VGG by a large margin. These experiments further reveal that the zero-padding strategy plays an important role in a position-dependent task, an observation that has been long-ignored in neural network solutions to vision problems.

| | Model | PASCAL-S | | BLACK | | WHITE | | NOISE | |
|---|---|---|---|---|---|---|---|---|---|
| | | SPC | MAE | SPC | MAE | SPC | MAE | SPC | MAE |
| **H** | VGG | .742 | .149 | .751 | .164 | .873 | .157 | .591 | .173 |
| | VGG-SOD | .969 | .055 | .857 | .099 | .938 | .087 | .965 | .060 |
| | VGG-SS | .982 | .038 | .990 | .030 | .985 | .032 | .985 | .033 |
| **G** | VGG | .814 | .109 | .842 | .123 | .898 | .116 | .762 | .129 |
| | VGG-SOD | .948 | .067 | .904 | .086 | .907 | .085 | .912 | .077 |
| | VGG-SS | .971 | .055 | .984 | .050 | .989 | .046 | .982 | .051 |
| **HS** | VGG | .405 | .556 | .532 | .583 | .576 | .574 | .375 | .573 |
| | VGG-SOD | .667 | .476 | .699 | .506 | .709 | .482 | .668 | .489 |
| | VGG-SS | .810 | .430 | .802 | .426 | .810 | .426 | .789 | .428 |

Table 6: Comparison of VGG models pretrained for classification, SOD, and semantic segmentation.

## 5 Conclusion

In this paper we explore the hypothesis that absolute position information is implicitly encoded in convolutional neural networks. Experiments reveal that positional information is available to a strong degree. More detailed experiments show that larger receptive fields or non-linear readout of positional information further augments the readout of absolute position, which is already very strong from a trivial single layer $3 \times 3$ PosENet. Experiments also reveal that this recovery is possible when no semantic cues are present and interference from semantic information suggests joint encoding of *what* (semantic features) and *where* (absolute position). Results point to zero padding and borders as an anchor from which spatial information is derived and eventually propagated over the whole image as spatial abstraction occurs. These results demonstrate a fundamental property of CNNs that was unknown to date, and for which much further exploration is warranted.

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
