# OpenReview forum: "How much Position Information Do Convolutional Neural Networks Encode?"
_ICLR.cc/2020/Conference — Accept (Spotlight)_

### Official Review · AnonReviewer2 · 2019-10-23
**Official Blind Review #2**

**Rating:** 8

**Review:**

The paper investigates to what degree Convolutional Neural Networks (CNNs) learn to encode positional information.
Rather interesting finding is the not only they do encode this information, but that it is to a large degree function of the padding commonly used in the CNN architectures.

The problem the paper is looking at is well motivated, the experiments are nicely designed and it includes comprehensive ablation study.
Previous and related work seems to be well referenced.
The main idea of introducing the PosENet to predict the gradient map is neat, and allows for interesting experiments (e.g. what layers most strongly encode the positional information).

I really enjoyed the paper, the overall quality is high and does not seem to be rushed (no obvious typos or mistakes in the figures/tables).
I believe this should be an accept.

Q:
I can understand why you removed the pooling layers, but did you try to run some of your experiments with these as well? How were the numbers effected?

**Experience Assessment:**

I have read many papers in this area.

**Review Assessment: Checking Correctness Of Derivations And Theory:**

I assessed the sensibility of the derivations and theory.

**Review Assessment: Checking Correctness Of Experiments:**

I assessed the sensibility of the experiments.

**Review Assessment: Thoroughness In Paper Reading:**

I read the paper at least twice and used my best judgement in assessing the paper.

---

> ### Author Response · Authors · 2019-11-15
> **Response to reviewer 2**
>
> We really appreciate your review and  we’re glad to hear you are pleased with the paper!
>
> Please let us further clarify the implementation details. We did not remove any pooling layers except the last average pooling layer in the ResNet, which was designed to compress the output in order to feed to a Fully Connected (FC) layer. The pooling layers within each network (convolutional part, sometimes called backbone) have been retained because the weight was trained based on that structure design. It is commonplace to replace the FC layers with conv layers as in most dense labeling tasks.

---

### Official Review · AnonReviewer1 · 2019-10-24
**Official Blind Review #1**

**Rating:** 8

**Review:**

This paper studied the problem of the encoded position information in convolution neural networks. The hypothesis is that CNN can implicitly learn to encode the position information. The author tests the hypothesis with lots of experiments to show how and where the position information is encoded.

Clarity:
This paper is interesting for me. It tries to understand the encoded position information that is easily ignored by researchers. I like adequate experiments with learned position information and position illustrations.

Experiments:
1. The paper mainly discussed the zero-padding and found it is the source of position information. How about other padding modes like constant-padding, reflection-padding, and replication-padding?

2. The partial convolution-based padding method [1] (padded regions are masked out) shows that its recognition accuracy is higher than the traditional zero-padding approach. Can you help investigate where the position information comes from for this case?

[1] Partial Convolution based Padding, https://arxiv.org/pdf/1811.11718.pdf.


Some of my concerns are well addressed by the author thus I upgrade my score.


**Experience Assessment:**

I have published one or two papers in this area.

**Review Assessment: Checking Correctness Of Derivations And Theory:**

I carefully checked the derivations and theory.

**Review Assessment: Checking Correctness Of Experiments:**

I carefully checked the experiments.

**Review Assessment: Thoroughness In Paper Reading:**

I read the paper thoroughly.

---

> ### Author Response · Authors · 2019-11-15
> **Response to reviewer 1**
>
> Many thanks for your review and we appreciate your insightful feedback.
>
> In our paper we discussed the implicit effect of the widely used zero-padding mechanism in CNNs. We believe the strong position information is encoded by the value transition near the boundary, zero to non-zero values. Intuitively, we believe other padding strategies, e.g. reflection or replication padding, are not able to deliver this clear position information.
>
> We compared the effect of Circular padding implemented in Pytorch with the commonly used zero-padding on the Horizontal (H) setting using VGG16, First row of Table 1 (VGG). The training loss of zero-padding starts from 0.045 and drops to 0.03 in the end. While the loss for circular-padding begins at 0.065 and ends at 0.056, much higher than zero-padding. The results of circular-padding on the PASCAL-S dataset are (SPC 0.381, MAE 0.224). Note that this result is similar to the setting of VGG w/o padding, Table 4 (VGG w/o padding on H). This further validates our hypothesis that the position information is delivered by the value transition of zero-padding.
>
> For the conv-padding paper, according to Equations (4) and (5), their method essentially still applies zero-padding, which means the position information should be encoded. Their method is actually weighing the output of the convolution based on how many zeros are padded, r(i, j).

---

### Official Review · AnonReviewer3 · 2019-10-24
**Official Blind Review #3**

**Rating:** 8

**Review:**

This paper studies whether and how position information is encoded in CNNs. On top of VGG and ResNet, it constructs an additional PosENet to recover position information. By analyzing how well PosENet recovers position information, this paper provides several interesting findings: CNNs indeeds encode position information and zero-padding is surprisingly important here.

[Pros]

1. I enjoy reading this paper: probing CNNs is not easy, but it designs experiments in an intuitive way and rigorously performs ablation studies and analysis.
2. The observations and findings are interesting and helpful to the community.

[Cons]

1. A weakness of this paper is that it ignores the impact of training process while probing PosENet: In Table 1, VGG/ResNet perform much better than PosENet, but it could be because VGG/ResNet is easier to train (kind of fine-tuning PosENet only) than PosENet. Would be nice to show the training curve and train PosENet longer.
2. Zero-padding seems to play a surprisingly important role in encoding position information (Table 5), but it is still unclear why it is so important and how it helps.

Overall, I think this is a good paper.


**Experience Assessment:**

I have read many papers in this area.

**Review Assessment: Checking Correctness Of Derivations And Theory:**

I assessed the sensibility of the derivations and theory.

**Review Assessment: Checking Correctness Of Experiments:**

I assessed the sensibility of the experiments.

**Review Assessment: Thoroughness In Paper Reading:**

I read the paper at least twice and used my best judgement in assessing the paper.

---

> ### Author Response · Authors · 2019-11-15
> **Response to reviewer 3**
>
> We thank Reviewer 3 for the detailed feedback and we will further explain the question raised in the comment.
>
> We think the first question is about initialization, (cold or hot start). We also thought about a longer training procedure for the PosENet because it was trained from scratch. But we found that the training loss does not decrease after the first several iterations, the weight becomes saturated quickly. The training loss of the PosENet converges at 0.084 after the first epoch. Also, all the test losses (on natural images PASCAL-S or synthetic images BLACK, WHITE or NOISE) are the same as the training loss. This suggests that the prediction may be completely independent of the content of images.
>
> We believe the reason behind this is that zero-padding delivers obvious boundary information, the transition between the zeros padded and the content. As discussed in the answer to Reviewer 1, we believe not all padding strategies can deliver this position information.

---

### Public Comment · ~Alberto_Bernacchia1 · 2020-01-08
**Clarification**

Hello, this is my first comment on OpenReview so please let me know if this not the right place for asking clarifications about a manuscript. I read this paper and I feel there is something fundamental I do not understand about it.

If I understood correctly, the input to the Position Encoding Module (PEM) is the combined outputs of a few layers of a pre-trained CNN, and the target output of the PEM is a fixed image (the value of x coordinates, for example). The target output image is always the same, fixed, irrespective of what's the input image. Then, I do not understand how we can draw any conclusions from this model.

The mutual information between the input and the output is zero by construction, and I can always set up a toy model that gives me the target output, irrespective of the input. For example, let say that my target output is a picture of my dog, always the same picture, irrespective of the input. I set up a toy model that discards the input and gives me always that output picture of my dog. Can I conclude that the input, whatever that was, has information about my dog?

---

> ### Author Response · Authors · 2020-01-09
> **Answers to Alberto Bernacchia**
>
> Dear Alberto,
>
> Thanks for this interesting question and we will further explain our work in this post.
>
> 1. I think your understanding about the network architecture is correct, PEM is designed to output a fixed image(the gradient groundtruth) based on the output of a pre-trained model, e.g., VGG or ResNet.
>
> 2. Can this model output a fixed image(say a dog) regardless the input? The short answer is it is possible to generate a fixed output, in this case, the learning process should be independent of position information(zero-padding). However, our experiment (Table 5) shows the output is indeed based on the position information delivered by zero-padding. An addition to your question, I personlly believe a model(given enough complexity) could blindly memorize each of the training image to output the desired prediction(say a dog), but I don't think this can give you the output you want on the test set.(https://arxiv.org/abs/1611.03530, ICLR2017).
>
> Regards
> Authors of paper 781

---

> > ### Public Comment · ~Alberto_Bernacchia1 · 2020-01-29
> > **Answer to author 781**
> >
> > Thank you for your answers.
> > Sorry I've just seen your response now! I thought I would get a notification by email but I didn't, so I thought you did not answer.
> > Concerning your point 2:
> > - I agree with you that a model with enough complexity can output whatever fixed image. However, it will not do so by memorizing the input, instead it will do so by ignoring the input. Therefore, it will work just fine on the test set.
> > - I understand that, in your experiment, the performance of PEN depends on zero padding. However, my opinion is that this dependency is a consequence of the architecture of PEM, which is completely arbitrary. My prediction is that, if you change the architecture of PEM, then you will find that its performance will not depend on zero padding any more, instead it will depend on some other parameter of the CNN. Since the choice of PEM is arbitrary, the dependency of its accuracy on whatever parameter of the CNN is meaningless.
> > - Let me give you a very simple example to prove my point.
> > Let say you have a scalar linear model:
> > y = w*x
> > where x is the input, y is the output and w is the weight, each one is a scalar number.
> > The goal is to output always the same number, let say this number is equal to z, regardless of the input, and we use a square loss function L = (y - z)^2.
> > The solution to this problem is
> > w = z * m / (s^2 + m^2)
> > where m and s^2 are the sample mean and variance of the input x.
> > The loss at the optimum is
> > L = z^2 * (1 - m^2 / (s^2 + m^2))
> > So the solution depends on the input (through its mean and variance), as it is the case in your experiments.
> > However, I can change the model by constraining W in some arbitrary way.
> > For example, let say z>0 and m>0, and  I constrain the weight to be w<0.
> > Then the solution is always w=0, and the loss is L = z^2, , regardless of the input.
> > You can explore other arbitrary ways of constraining w, and you will find different dependencies of the loss on the input. Since your choice of the constraints are arbitrary, so are those dependencies.

---

> > > ### Author Response · Authors · 2020-01-29
> > > **Answers to Alberto Bernacchia**
> > >
> > > Thanks for your further question. We thought about when the model is input-agnostic, w=0. In that case, the model is not able output the pre-define pattern. We did not experience this problem of zero vector in our experiment as well. You might want to check the answer to Reviewer 1, in which we showed the effect of circular padding. If somehow the model was adjusted to "z * m / (s^2 + m^2)", the circular padding should improve the performance. Please correct me if I am wrong and help me recall other cases that the model can produce this output without knowing the position information.
> > >
> > > Regards
> > > Sen

---

> > > > ### Author Response · Authors · 2020-01-29
> > > > **Answers to Alberto Bernacchia**
> > > >
> > > > In addition, the scores on the synthetic images, white, black and noise in Table 1 could be a proof to question the hypothesis "w = z * m / (s^2 + m^2)".

---

> > > > > ### Public Comment · ~Alberto_Bernacchia1 · 2020-02-21
> > > > > **Answers to Paper781 Authors**
> > > > >
> > > > > Thank you for your response, i understand that the PEM is unable to output the pre-defined pattern for w=0.
> > > > >
> > > > > I have another two questions, which I mentioned already in comments above.
> > > > > 1) Given that the point of the paper is to analyze the information inside a CNN, do you agree that the choice of the architecture of the PEG is arbitrary? In other words, if I repeat your study, but I use a different architecture for the PEG, then my results are equally valid.
> > > > > 2) If you agree with 1, then I choose a PEG that ignores the input, and always gives the pre-defined output. The fact that my input is the activity taken from a CNN does not matter, it can be anything. Then, according to your logic, I conclude that everything (every possible input) has information about the pre-defined output, which is obviously wrong. In fact, in that case the mutual information between the input and the output of the PEG is zero.

---

> > > > > > ### Author Response · Authors · 2020-02-21
> > > > > > **Answers to Alberto Bernacchia**
> > > > > >
> > > > > > Dear Alberto Bernacchia:
> > > > > >
> > > > > > Thanks for your new questions.
> > > > > > 1) "analyze the information inside a CNN, do you agree that the choice of the architecture of the PEG is arbitrary?".
> > > > > > A: Yes, we analyzed the information inside a CNN about the position information. The choice of PEM could be arbitrary but note that this PEM is only a read-out module. You can replace it with one or more conv layers and your result should be valid.
> > > > > >
> > > > > > 2)  "then I choose a PEG that ignores the input, and always gives the pre-defined output. The fact that my input is the activity taken from a CNN does not matter, it can be anything."
> > > > > > A: I assume you are trying to put a PEM on top of the input data directly. Note that we mentioned the source of the position information is from zero-padding, you can have the position information if you put the "source" there.
> > > > > >
> > > > > > In a way you can consider the backbone CNN(resnet or vgg) does not matter. On the contrary, however, we were trying to explore if a widely used backbone (resnet or vgg) contains this position information and where does it come from, we "interpret" our work as CNN understanding. Because this long-ignored problem is important in computer vision tasks, given that the absolute position can be converted to relative position in theory.
> > > > > >
> > > > > > Thanks
> > > > > > Sen

---

> > > > > > > ### Author Response · Authors · 2020-02-21
> > > > > > > **Answers to Alberto Bernacchia**
> > > > > > >
> > > > > > > Addition to my answer, our work was trying to analyze and understand the backbone by using PEM, instead of designing a PEM to output position information.
> > > > > > >
> > > > > > > Cheers
> > > > > > > Sen

---

### Public Comment · ~Thomas_Brox1 · 2020-06-24
**positional encoding vs. constant features**

Are you sure that the results tell something about positional encoding in the feature embedding? All the PEN needs is a feature channel that is constant for all inputs (like a bias). Then the PEN can normalize this constant to 1 and multiply it with the desired constant output.

---

> ### Author Response · Authors · 2020-06-29
> **Answers to Thomas Brox**
>
> Hi Thomas Brox, thanks for your interesting question. I would like to discuss about this question, but can I ask if you are asking the input of the PEN module is a constant value? The input of the PEN is directly extracted from a pre-trained CNN model, so are you asking if given a natural image X, the multi-level feature from a VGG could be a constant? I can better answer this question if understand it correctly, thanks Sen.

---

### Public Comment · ~Mai_Nguyen1 · 2022-12-14
**Different position map, same input?**

How can the PosENet based network output different position map while giving the same input? (Fig 4). Isn't the weight of the network fixed after training, then we can expect the network to get same position giving same input no?

---

### Decision · Program_Chairs · 2019-12-19

**Decision:**

Accept (Spotlight)

**Comment:**

This paper analyzes the weights associated with filters in CNNs and finds that they encode positional information (i.e. near the edges of the image).  A detailed discussion and analysis is performed, which shows where this positional information comes from.

The reviewers were happy with your paper and found it to be quite interesting.  The reviewers felt your paper addressed an important (and surprising!) issue not previously recognized in CNNs.